# Cyclosporin A as a Source for a Novel Insecticidal Product for Controlling *Spodoptera frugiperda*

**DOI:** 10.3390/toxins14100721

**Published:** 2022-10-21

**Authors:** Chengxian Sun, Shunjia Li, Kai Wang, Hongqiang Feng, Caihong Tian, Xiaoguang Liu, Xiang Li, Xinming Yin, Yanmei Wang, Jizhen Wei, Shiheng An

**Affiliations:** 1State Key Laboratory of Wheat and Maize Crop Science, Henan International Laboratory for Green Pest Control, College of Plant Protection, Henan Agricultural University, Zhengzhou 450002, China; 2Henan Academy of Agricultural Sciences, Zhengzhou 450002, China; 3College of Forestry, Henan Agricultural University, Zhengzhou 450002, China

**Keywords:** CsA, combined toxicity, insecticidal activity, *Spodoptera frugiperda*, sublethal effect

## Abstract

The fall armyworm (FAW), *Spodoptera frugiperda*, causes substantial annual agricultural production losses worldwide due to its resistance to many insecticides. Therefore, new insecticides are urgently needed to more effectively control FAW. Cyclosporin A (CsA) is a secondary metabolite of fungi; little is known about its insecticidal activity, especially for the control of FAW. In this study, we demonstrate that CsA shows excellent insecticidal activity (LC_50_ = 9.69 μg/g) against FAW through significant suppression of calcineurin (CaN) activity, which is a new target for pest control. Combinations of CsA and indoxacarb, emamectin benzoate, or Vip3Aa showed independent or synergistic toxicity against FAW; however, the combination of CsA and chlorantraniliprole showed no toxicity. Sublethal doses of CsA led to decreases in FAW larval and pupal weight, pupation, emergence, mating rates, adult longevity, extended development of FAW larvae and pupae and the pre-oviposition period of adults, and increases in the proportion of pupal malformation. Importantly, CsA treatment reduced FAW ovarian size and female fecundity, which suggests that it has great potential to suppress FAW colony formation. Taken together, these results indicate that CsA has high potential as an insecticide for controlling FAW.

## 1. Introduction

Cyclosporin A (CsA; molecular formula: C_62_H_111_N_11_O_12_, MeBmt1-Abu2-Sar3-MeLeu4-Val5-MeLeu6-Ala7-d-Ala8-MeLeu9-MeLeu10-MeVal11) is a fungal metabolite that is produced by many fungi, such as *Tolypocladium*
*inflatum* [1]. This metabolite exhibits strong immunosuppressive activity and is widely used in organ transplantation and the treatment of various autoimmune diseases [1,2]. CsA is a specific inhibitor of calcineurin (CaN), which is a Ca^2+^-activated enzyme that plays a key role in immune responses. Once CsA enters the cell, it is bound by cyclophilin, an immunophilin that suppresses CaN activity and prevents dephosphorylation of the nuclear factor of activated T-cells (NFAT), leading to the suppression of immunoreaction by T-helper cell proliferation, and ultimately causing a severe reduction in T-suppressor cells [3].

To address global food shortages, novel high-efficiency and low-toxicity insecticides are being explored to improve agricultural yield by reducing pest damage to crops. CsA has great potential for insect management due to its insecticidal activity in autogenous *Culex pipiens* Linnaeus (Diptera: Culicidae) larvae [4]. Treatment of *Galleria mellonella* Linnaeus (Lepidoptera: Pyralidae) larvae with CsA increases larval mortality via infection with the insect bacterial pathogen *Pseudomonas aeruginosa*, because CsA suppresses the humoral immune response of the insect [5]. A lower concentration of CsA increases mortality in the cattle tick *Rhipicephalus microplus* Canestrini (Parasitiformes: Ixodidae) when exposed to ivermectin, demonstrating the potential of CsA for pest control [6]. However, naturally produced CsA has not been studied as an insecticide for controlling agricultural pests.

The fall armyworm (FAW), *Spodoptera frugiperda* (J. E. Smith) (Lepidoptera: Noctuidae), is one of the most destructive crop pests worldwide, causing substantial economic losses. FAW larvae feed on more than 350 plants across 76 families, including many commercial crops such as *Zea mays*, *Sorghum bicolor*, *Oryza sativa*, and *Triticum aestivum* [7]. FAW can cause a 70% drop in maize yield when maize plants are attacked during the early stages of their development [8]. In sub-Saharan Africa, FAW has devastating impacts on the output of maize, rice, sorghum, and sugarcane, resulting in a loss of USD 13 billion each year [9]. At present, FAW is considered a global pest [7].

Over time, FAW has developed resistance to more than 40 chemical insecticides (such as organophosphorus, pyrethroid, bisamide, and spinosad) and biogenic insecticides, such as *Bacillus thuringiensis* (*Bt*, including Cry1Ab, Cry1F, Cry2Ab2, Cry1A.105, and Vip3Aa) [10,11,12,13,14]. As a result, FAW management still depends on chemical insecticides, where increasing insecticide dosage requirements maintain a vicious cycle of growing insecticide resistance, increasing pesticide residues, and greater harm to natural enemies [7,15]. Therefore, new insecticides with new action mechanisms must be explored.

In this study, we tested the hypothesis that the insecticidal activity of CsA will be effective against FAW, as well as testing its sublethal effects and combined toxicity with other insecticides. We also explored the insecticidal mechanism of CsA against FAW larvae. Our results demonstrate that CsA could be used as an insecticide to control this Lepidopteran pest with a new mechanism of action.

## 2. Results

### 2.1. Insecticidal Activity of CsA on FAW

Mortalities were determined in newly hatched and third instar FAW larvae fed with CsA for seven days (Table 1). The 50% and 95% lethal concentrations (LC_50_ and LC_95_, respectively) of CsA against neonates were 9.69 μg/g and 82.96 μg/g, respectively. The LC_50_ and LC_95_ values of CsA against third instar larvae were 260.13 μg/g and 1997.89 μg/g, respectively.

### 2.2. CsA Inhibits CaN (Insecticidal Target) Activity of Larvae

The results showed that all tested treatments of CsA significantly suppressed CaN activity at different time points. Relative CaN activities decreased by 32.65%, 38.45%, and 55.66% on the first day of CsA treatment with 25 µg/g, 50 µg/g, and 100 µg/g CsA, respectively (F = 13.04, df = 11, *p* = 0.0019); on the third day of CsA treatment, CaN activities decreased by 33.98%, 59.64%, and 78.16%, respectively (F = 31.90, df = 11, *p* = 0.0001); on the fifth day of CsA treatment, CaN activities decreased by 59.12%, 76.75%, and 79.36%, respectively (F = 57.20, df = 11, *p* = 0.0001); and on the seventh day of CsA treatment, CaN activities decreased by 40.58%, 66.21%, and 69.32%, respectively (F = 86.87, df = 11, *p* = 0.0001) (Figure 1).

### 2.3. CsA Toxicity in Third Instar Larvae when Treatment Combined with other Insecticides

Results showed that all combinations of indoxacarb and CsA exhibited synergistic toxicity in third instar larvae after treatment for three days (Figure 2A,B). However, most combinations of chlorantraniliprole and CsA showed antagonistic toxicity in third instar larvae after treatment for three days, except for 1 ng/g chlorantraniliprole and 50/100 μg/g CsA (which showed independent effects) (Figure 2C,D). Next, the combined toxicity of emamectin benzoate/Vip3Aa and CsA was tested after treatment for seven days. Independent effects were observed for all emamectin benzoate and CsA treatments except for 20 ng/g emamectin benzoate and 200 CsA, which showed a synergistic effect (Figure 2E,F). Among the combinations of Vip3Aa and CsA, both 0.8 μg/g Vip3Aa and 50/200 μg/g CsA and 1.6 μg/g Vip3Aa and 200 μg/g CsA showed independent toxic effects, while 1.6 μg/g Vip3Aa and 50/100 μg/g CsA showed significant synergistic toxicity. However, antagonistic toxic effects were observed for the 0.8 μg/g Vip3Aa and 100 μg/g CsA treatment (Figure 2G,H).

### 2.4. Sublethal Effects of CsA on FAW Larvae

The effects of CsA on FAW were investigated at final concentrations of 0 μg/g, 25 μg/g (LC_3_), 50 μg/g (LC_9_), and 100 μg/g (LC_23_). The results revealed that CsA treatment significantly arrested FAW development, compared with the control (Figure 3A). After eight days of treatment, partially developed larvae under the control treatment had developed to the prepupal stage; however, no prepupae were found among the CsA-treated populations, and the proportion of fifth and sixth instar larvae gradually decreased as the CsA concentration increased (Figure 3B). The average development period of the larvae (third instar to pupal stage) increased from 12.61 to 16.53 days (F = 68.76; df = 331; *p* = 0.0001; Figure 3C). Larval weight was significantly reduced after CsA treatment for two, four, six, and eight days (Appendix A).

### 2.5. Sublethal Effects of CsA on Pupae

Treating with CsA dramatically reduced the pupation rate from 83.06% (control) to 38.61% (100 μg/g CsA) (F = 12.68, df = 11, *p* = 0.0021; Figure 3A). No significant sex difference was observed in any of the treatment groups; however, the body size and weight of female and male pupae decreased significantly as the CsA doses increased (Appendix A and Figure 4B). As was observed in larvae, CsA treatment also delayed pupal development (Appendix A). Finally, malformed pupae with larval heads were observed in the CsA treatment groups (Figure 4C). CsA treatment led to abnormal pupation. The percentages of larvae that did not pupate, or that rarely pupated, were 0.65%, 7.79%, 27.92%, and 54.71% at 0 μg/g, 25 μg/g, 50 μg/g and 100 μg/g CsA concentrations, respectively (F = 65.05, df = 11, *p* = 0.0001) (Figure 4D), suggesting that high CsA doses led to high deformity rates. Further analysis revealed no significant differences in the malformation ratio between female and male pupae in the CsA-treated group (Appendix A).

### 2.6. Sublethal Effects of CsA on Adults

CsA treatment significantly decreased the emergence rates of adults from 85.43 ± 1.81% (control) to 56.82 ± 0.44% (100 μg/g CsA) (F = 28.55, df = 11, *p* = 0.0001) (Figure 5A). However, the emergence rates of males and females did not significantly differ in any of the treatment groups (Appendix A). As a result, the female/male ratios of adults were similar under different concentrations (Appendix A). CsA treatment also reduced adult body size (Figure 5B). Further experiments demonstrated that the mean longevities of female adults were longer in the control than those in the 25 μg/g and 50 μg/g CsA-treated groups (F = 5.93, df = 174, *p* = 0.0007) (Appendix A). However, only the 50 μg/g CsA treatment reduced the longevity of male adults (F = 4.15, df = 173, *p* = 0.0072; Appendix A).

### 2.7. Sublethal Effects of CsA on Adult Reproduction and Egg Hatching

Sublethal doses of CsA also had negative effects on adult reproduction and egg hatching. CsA treatment significantly suppressed the ovary development of adults compared with those in the control group (Figure 5C). Oviduct length decreased from 4.47 cm (control) to 2.83 cm (100 μg/g CsA) (F = 56.18, df = 148, *p* = 0.0001; Figure 5D), and the number of mature eggs in the ovaries decreased from 233.35 (control) to 25.52 (100 μg/g CsA) (F = 22.05, df = 148, *p* = 0.0001; Appendix A). The mating rate of adults also gradually decreased from 69.95% to 22.84% as the CsA concentrations increased (F = 24.47, df = 11, *p* = 0.0002; Figure 5E). CsA treatment extended the preoviposition period (F = 19.63, df = 78, *p* = 0.0001; Appendix A), and the total number of eggs laid by females per treatment decreased from 998.75 (control) to 175.25 (100 μg/g CsA) (F = 32.16, df =78, *p* = 0.0001; Figure 5F). As expected, egg hatching was also inhibited by CsA, and the hatching rate decreased from 85.72% to 31.04% (F = 66.05, df = 78, *p* = 0.0001; Figure 5G). The food intake of CsA-treated larvae was lower than that of control larvae. In addition, the change curves for pupal weight, ovary length, and number of eggs laid by adults in response to CsA treatment were analyzed. It was found that the slopes for ovary length, number of eggs per ovary, and total egg production in response to CsA treatment were lower than the slope for pupal weight (Appendix A), indicating that reproductive outcomes of CsA-treated FAW were influenced solely by nutrition.

## 3. Discussion

CsA is a biogenic insecticide that is produced by fungi [16]. The toxicity of CsA against FAW neonates after seven days was similar to that of *Bt* toxic proteins such as Cry1Da_7 and Cry2Ab2, which have mean LC_50_ values of 8.3 (4.5–16.7) μg/g and 11.81 (9.98–14.21) μg/g, respectively [17,18]. However, third instar larvae exhibited low susceptibility to CsA, and this phenomenon was also observed in larvae fed with corn leaves expressing Cry1Ab or Cry1Ab and Vip3Aa [19]. The toxicity of CsA against third instar FAW larvae was similar to that of Cry1Ac, each with LC_50_ values > 100 μg/g. Cry1Ac significantly suppressed the development of FAW larvae [20], and similarly, sublethal CsA not only suppressed the development of FAW larvae, but also reduced the FAW population. This result has important implications for integrative pest management strategies.

We explored the mechanism of action of CsA on FAW larvae; CsA had a significant inhibitory effect on CaN activity, consistent with its function as a specific enzyme inhibitor of CaN [3]. CaN is an important immune-regulated enzyme that regulates the expression of antimicrobial peptides and immune-related genes in insects through Toll or IMD pathways [21,22,23]. Suppression of CaN activity by FK506 increased the toxicity of Cry1Ac and Cry2Ab in *Helicoverpa armigera* Hubner (Lepidoptera: Noctuidae) larvae in our previous studies [24,25]. CaN is an important Ca^2+^-dependent phosphatase in animals, and it has a high amino acid identity among different species, indicating the conservation of CaN function [26,27]. Therefore, it appears that CsA may inhibit the CaN-regulated pathway to exert its insecticidal effects. These results indicate that CaN could be a new target for pest control.

It is very important to evaluate the toxicity of insecticides to humans and beneficial insects, even though CsA was approved by the Food and Drug Administration in 1983 [28]. Recent studies have also confirmed that CsA is safe for humans, and even for pregnant women [29]. CsA doses used in clinical applications are from 3 mg/kg to 25 mg/kg per day [30], which may be greater than the dose used for insect control. Nevertheless, exposure to CsA would be different for humans through insecticide application (inhalation or contact) versus clinic application (ingestion), and therefore the effects of residual CsA on humans are not clear. Beneficial insects, like wasp parasitoids, may benefit from CsA application and increase in population because the parasitic ability of wasp parasitoids is closely related to the immune response of host insects [31,32]. The eggs and larvae of wasp parasitoids may survive more easily in host insects whose immunoreaction has been inhibited by CsA. Therefore, further toxicological tests of CsA are required for humans, mammals, and other non-target species.

Tank mixes and pre-packs are combinations of two or more pesticides applied as a single mixture to improve pesticide effectiveness. In this study, the independent or synergistic toxic effects on FAW larvae of various combinations of CsA with indoxacarb, emamectin benzoate, or Vip3Aa indicated the potential of CsA in pest control strategies based on tank mixes or pre-packs. Importantly, combinations of pesticides with different modes of action could prevent or reduce insecticide resistance. Among the insecticides that were combined with CsA in this study, indoxacarb is bioactivated by esterase to prevent the entrance of Na^+^ into nerve cells and block nerve conduction [33]. Emamectin benzoate causes the indraft of Cl^-^, which is an allosteric modulator of the γ-aminobutyric acid receptor (GABA) and glutamate-gated chloride channel (Glu-Cl) in nerve cells, and which prevents muscle contractions [34]. Unlike indoxacarb and emamectin benzoate, CsA binds to cyclophilin and inhibits CaN activity by increasing Ca^2+^ concentrations after CsA enters the cells [3]. This action model may cause synergistic combined toxicity with indoxacarb/emamectin benzoate and CsA, as the intracellular Ca^2+^ rise not only renders sodium channels more sensitive to indoxacarb, but also activates chloride channels [35,36]. Additionally, the suppression of immunoreaction caused by CsA may increase the amount of midgut microbes, which may enhance the toxicity of Vip3Aa against FAW larvae. Vip3Aa mainly acts on the receptors of brush border membrane vesicles, resulting in the swelling and lysing of midgut epithelium cells [37], which may allow more midgut microbes enter into the hemolymph and lead to more larvae dying from septicemia. Our results indicate that CsA may be used in integrated resistance management. However, the combined effects of CsA and chlorantraniliprole against FAW larvae were found to be antagonistic or independent. This may cause different effects on Ca^2+^ concentrations. An increase in Ca^2+^ concentration is required for CaN inhibition of CsA [3], but this process may blunt the effect of chlorantraniliprole on insects, because chlorantraniliprole activates ryanodine receptors, thereby inducing muscle cells to excrete Ca^2+^ [38]. Thus, the use of CsA and chlorantraniliprole, as well as CsA in combination with chlorantraniliprole-like insecticides should be avoided in pest control.

In addition to its insecticidal activity against FAW larvae, we investigated the adverse effects of sublethal CsA on FAW. Beyond the adverse effects of CsA on the development of larvae, pupae, and adults, which are common with other insecticides, some interesting phenomena were observed. For example, the proportion of malformed pupae was higher in CsA-treated group than in the control group. This was likely due to abnormal titers of juvenile hormone (JH) and 20-hydroxyecdysone (20E), which regulate insect metamorphosis [39]. In Lepidoptera, a high 20E titer promotes pupation, whereas a high JH titer prevents larval development [39,40]. Abnormal pupae with larval characteristics were also found among *H. armigera* exposed to spinosad, as seen in JH and 20E hormone disorders [41]. Therefore, CsA may interfere with JH and 20E “pathways”, thus affecting insect metamorphosis.

CsA also negatively affected adult reproduction, including ovary size, mating, preoviposition, fecundity, and hatching ratio. Considering the fact that larval/pupal body weights decreased significantly in the CsA-treated groups, nutrient deficiency may explain these results. Indeed, many insecticides also suppress feeding behavior and digestive enzyme activity, leading to nutrient deficiency, and ultimately hindering reproductive development [42,43]. However, the adverse effects of CsA on adults appeared to differ from those of other insecticides: the decreases in the ovary length and egg production were significantly greater than the decrease in pupal weight as the CsA dose increased. Sex pheromone-induced mating is an important process in Lepidopteran insect reproduction. The inhibition of sex pheromone biosynthesis significantly reduces mating and fecundity in adults. The regulation of sex pheromone biosynthesis is conserved in females, and CaN is the dominant enzyme in Lepidopteran species that regulates this biological process [44]. The production of sex pheromones in *H. armigera* and *Ostrinia furnacalis* Guenée (Lepidoptera: Pyralidae) was significantly reduced after the suppression of CaN activity [45,46]. Additionally, CaN is required for both male sex pheromone biosynthesis and female acceptance [27]. Therefore, accumulated CsA may decrease sex pheromone production, ultimately reducing the mating rate. In addition, CaN promotes the formation of insect myofilaments, thereby affecting flight ability [47], which may further suppress mating. CaN is essential for the completion of female meiosis in *Drosophila melanogaster* Meigen (Diptera: Drosophilidae), and for normal egg development [48,49]. Similarly, egg hatching rates in our CsA-treated groups were significantly lower than in the control group in our study, possibly due to the inhibition of CaN activity by CsA. Given the ovarian development accompanying oogenesis [48], we cannot rule out an effect of CsA. This merits further investigation.

## 4. Conclusions

In this study, the lethal and sublethal effects of CsA as a potential insecticide against FAW were investigated. The results demonstrate that CsA shows excellent insecticidal activity against newly hatched and third instar FAW larvae. Moreover, CsA showed good toxicity in combination with indoxacarb, emamectin benzoate, and Vip3A. Sublethal concentrations of CsA negatively affected FAW development and reproduction by inhibiting CaN activity. The physiological and morphological changes induced by sublethal CsA may provide insight into the growth, development, pupation regulation, ovarian development, and immune mechanisms of FAW. Understanding these mechanisms and pathways will allow us to determine the potential of CsA as a new insecticide, and lead to the identification of new insecticidal targets.

## 5. Materials and Methods

### 5.1. Insects

The FAW population was derived from specimens collected in Ruili, Yunnan Province, China. This population was reared on a diet for over 10 generations without exposure to any pesticides, following the methods described in the Appendix A.

### 5.2. CsA and Insecticides

CsA (98.5% active ingredient) was purchased from Beijing Solarbio Science and Technology Co., Ltd. (Beijing, China). Indoxacarb (99% active ingredient), chlorantraniliprole (95% active ingredient), and emamectin benzoate (91% active ingredient) were kindly provided by Dr. Bin Zhu (Department of Entomology, China Agricultural University, Beijing, China). Vip3Aa protein was purchased from Beijing Genralpest Biotech Research Co., Ltd. (Beijing, China).

### 5.3. Bioassay of CsA Insecticidal Activity

To determine the insecticidal activity of CsA against FAW, different doses of CsA were added to the artificial diet of the FAW at a temperature of 42 °C. The final concentrations of CsA in the diet were 0 μg/g, 1.56 μg/g, 3.13 μg/g, 6.25 μg/g, 12.5 μg/g, 25 μg/g, or 50 μg/g for newly hatched larvae, and 0 μg/g, 25 μg/g, 50 μg/g, 100 μg/g, 200 μg/g, 400 μg/g, or 800 μg/g for third instar larvae. The content of dimethyl sulfoxide (DMSO), a CsA solvent, in the artificial diet was measured to within 0.1%; a diet supplemented with DMSO was used as the control. Newly hatched larvae or newly molted third instar larvae of the same size were provided with approximately 0.5 g per day of fresh artificial diet containing CsA in a 25-mL cup. We used 120 larvae for each treatment, with three biological replicates. At seven days after CsA exposure, larval mortality was recorded. Larvae who exhibited no response to physical stimulation and larvae whose instar remained unchanged after seven days were recorded as dead.

### 5.4. CaN Activity Measurement

Newly molted third instar larvae were orally exposed to 0 µg/g, 25 µg/g, 50 µg/g, and 100 µg/g CsA for one, three, five, and seven days, respectively, and subsequently subjected to midgut dissection followed by measurement of the CaN activity. The midgut was dissected, fully ground in 0.7% saline solution on ice, and subjected to protein extraction. After centrifugation at 1500× *g* for 10 min at 4 °C, the supernatant was collected, and protein concentrations were determined using a BCA Protein Assay Kit (Beyotime Biotechnology, Shanghai, China), following the manufacturer’s instructions. A Calcineurin Activity Assay Kit (Abcam, Cambridge, UK) was used for the CaN activity assay, according to the manufacturer’s instructions [50]. Midgut proteins and CaN assay buffer were mixed and a CaN substrate was then added and incubated for 10 min. Next, a color development reagent was added and the (OD) 636 nm was read by the SynergyH1 Hybrid Multi-Mode Reader (BioTek, Winooski, VT, USA). CaN activity was calculated following the formula provided by the manufacturer’s instructions. Each treatment had three replicates, and each replicate included at least eight midgut organs.

### 5.5. Toxicity Tests for the Combination Treatments (CsA and Other Insecticides)

We mixed four pesticides with CsA (50 μg/g, 100 μg/g, or 200 μg/g): indoxacarb (final concentration in artificial diet: 20 μg/g, 40 μg/g), chlorantraniliprole (0.5 ng/g or 1 ng/g), emamectin benzoate (20 ng/g or 40 ng/g), and the *Bt* protein toxin Vip3Aa (0.8 μg/g or 1.6 μg/g). These insecticides are described in detail in the Appendix A. Single-insecticide treatment was the control. These insecticides and their concentrations were selected according to an indoor bioassay against FAW performed in our laboratory (unpublished data). All concentrations of each insecticide led to a mortality rate < 36.11%. For each combination, we calculated the mortality of CsA (50 μg/g, 100 μg/g, and 200 μg/g) against third instar larvae. The expected mortality was calculated as described previously [24,25], and as explained in detail in the Appendix A. Each treatment group consisted of 72 newly molted third instar larvae of the same size (three biological replicates). Mortality was recorded daily until all larvae had died, or until seven days after treatment.

### 5.6. Bioassays of Sublethal Effects of CsA on FAW

According to the susceptible toxicity baseline of CsA against third instar FAW larvae (bioassay of CsA insecticidal activity), we selected 0 μg/g, 25 μg/g, 50 μg/g, or 100 μg/g (sublethal concentrations of LC_3_, LC_9_, or LC_23_) of CsA to test the sublethal effects. Newly molted third instar larvae of the same size (120 larvae for each replicate) were fed a fresh artificial diet including different concentrations of CsA (approximately 0.5 g, replaced daily) in a 25 mL cup until they developed to the prepupal stage. All adults were fed 10% sucrose solution without CsA or DMSO every day. Three replicates were conducted. The sublethal effects of CsA on larvae, pupae, and adults were recorded following methods described in the Supplementary Information.

### 5.7. Statistical Analysis

All data are presented as the mean ± standard error (SE). Mortality data were subjected to probit analysis using SPSS software (ver. 20.0; IBM Corp., Armonk, NY, USA). Heterogeneity was used in the calculation of confidence limits when *p* < 0.15. Significant differences of multiple comparisons were analyzed by variance (ANOVA), followed by a Tukey HSD test (*p* < 0.05) using SPSS (ver. 20.0) software. An independent samples *t*-test was performed for analysis of significant differences of paired comparisons using SPSS (ver. 20.0) software.

## Figures and Tables

**Figure 1 toxins-14-00721-f001:**
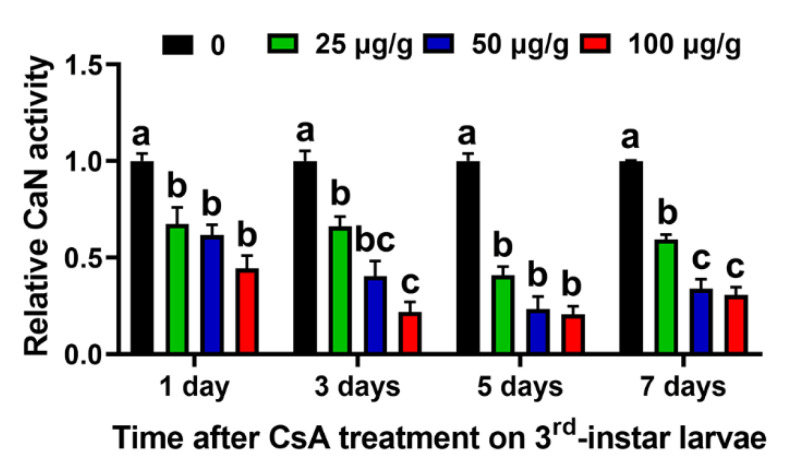
Inhibitory effects of different CsA concentrations on CaN activity in third instar larvae after one, three, five, and seven days. Data are means ± SE of three biological replicates. The different lowercase letters on the error bars indicate significant differences analyzed using ANOVA followed by a Tukey test at the level of *p* < 0.05.

**Figure 2 toxins-14-00721-f002:**
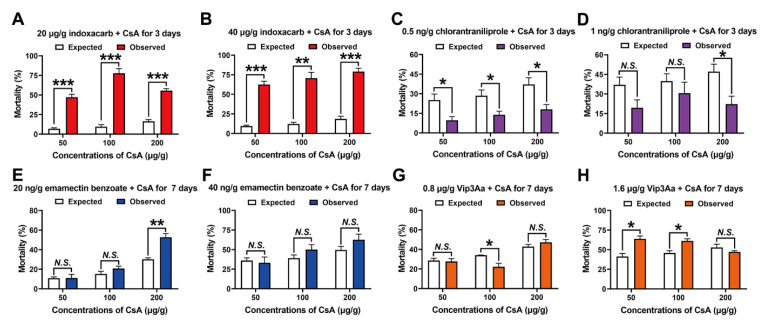
Combined toxicity of CsA and four insecticides against third instar larvae of FAW. (**A**,**B**) CsA with indoxacarb for three days. (**C**,**D**) CsA with chlorantraniliprole for three days. (**E**,**F**) Combined toxicity of CsA with emamectin benzoate for seven days. (**G**,**H**) CsA with Vip3Aa for seven days. All data are means ± SE of three biological replicates. The significant differences were analyzed by independent sample *t*-test. “***” means *p* < 0.001, “**” means *p* < 0.01, “*” means *p* < 0.05, and “*N.S.*” means *p* > 0.05.

**Figure 3 toxins-14-00721-f003:**
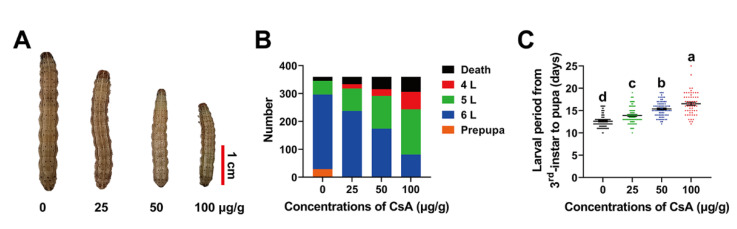
Effects of sublethal CsA on development of newly emerged third instar FAW larvae. (**A**) Phenotypes of larvae treated with 0 μg/g, 25 μg/g, 50 μg/g, and 100 μg/g CsA for eight days. (**B**) Number of larvae that stayed at different developmental stages after CsA treatment for eight days. (**C**) Effects of CsA on larval periods from third instar to pupa. Data of larval periods are means ± SE of 120 specimens. The significance of differences was analyzed using ANOVA followed by Tukey test software at the level of *p* < 0.05, and is marked with different lowercase letters. Note: 4 L, 5 L, and 6 L refer to fourth, fifth, and sixth instar larvae, respectively.

**Figure 4 toxins-14-00721-f004:**
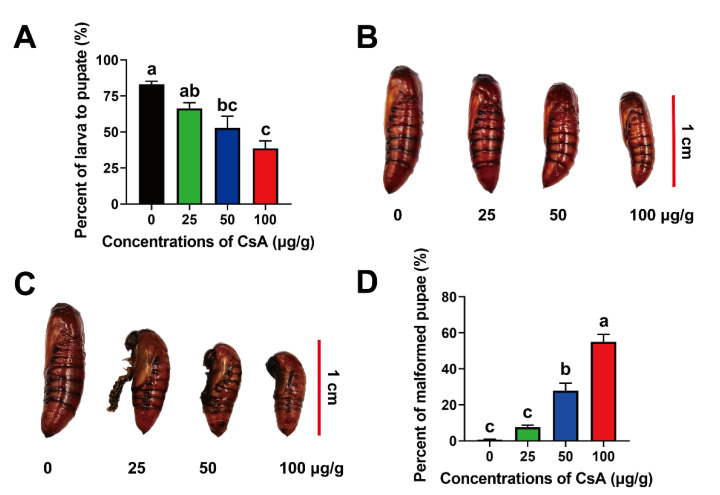
Effects of sublethal CsA on FAW pupae. (**A**) Percentages of larvae that pupated under the sublethal CsA treatments. (**B**) Effects of CsA on ratio of female/male pupae. (**B**) Phenotypes of pupae. (**C**) Malformed phenotypes of pupae caused by sublethal CsA. (**D**) Percentages of malformed pupae under different concentrations of CsA. Data are means ± SE of more than three biological replicates. The lowercase letters on the error bars indicate the significant differences at the level of *p* < 0.05, which were calculated using ANOVA followed by a Tukey test.

**Figure 5 toxins-14-00721-f005:**
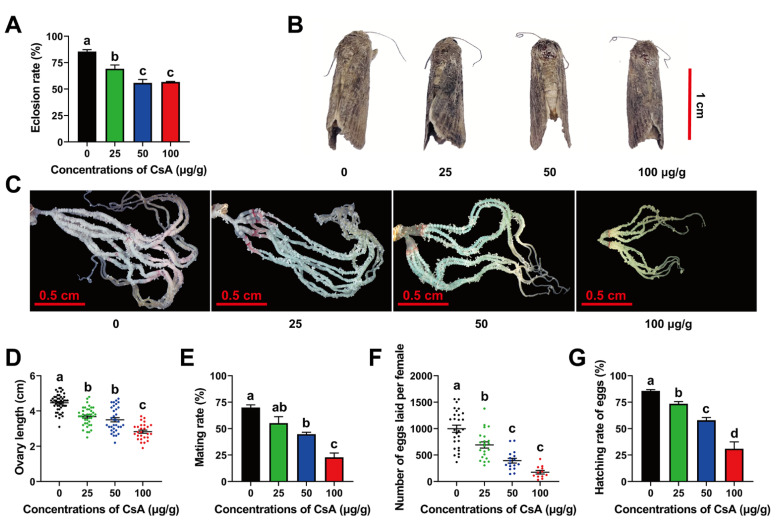
Effects of CsA on adult FAW, their reproduction, and their eggs. (**A**) Effects of sublethal CsA on eclosion rates. (**B**) Phenotypes of adults treated with CsA. (**C**) Ovarian phenotypes under CsA treatments. (**D**) Changes of ovary length caused by CsA. (**E**) Mating rates at different CsA concentrations. (**F**) Numbers of eggs laid by female adults in each group. (**G**) Hatching rates of eggs laid by female adults treated with different concentrations of CsA. At least three biological replicates were used to calculate the error bars displayed with means ± SE. The different lowercase letters on the error bars indicate significant differences analyzed using ANOVA and a Tukey test at the level of *p* < 0.05.

**Table 1 toxins-14-00721-t001:** Insecticidal activity of CsA against FAW larvae after treatment for seven days.

Larvae	LC_50_ (95% CL ^a^, μg/g)	LC_95_ (95% CL, μg/g)	Slope ± SE	χ^2^	df	*p* ^b^
First instar	9.69 (5.95–16.46)	82.96 (38.00–514.634)	1.76 ± 0.12	22.01	4	<0.01
Third instar	260.13 (162.98–490.31)	1997.89 (869.31–15,405.45)	1.86 ± 0.13	23.31	4	<0.01

^a^ CL: confidence limit. ^b^ Heterogeneity was used in the calculation of confidence limits when *p* < 0.15.

## Data Availability

The data presented in this study are available on request from the corresponding author.

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
