# Peer review of "Cyclosporin A as a Source for a Novel Insecticidal Product for Controlling *Spodoptera frugiperda"

_toxins, 2022, doi:10.3390/toxins14100721_

Round 1
Reviewer 1 Report
The paper titled "Cyclosporin A as a novel source of inscticidal product for controlling in Spodoptera frugiperda" presented a lot of results indicating CsA was toxic to FAW or inhibit the armyworms development.
1. As mentioned in the paper, the authors used high doses (100ug/g) of CsA to interfere FAW growth and development. Do authors consider the CsA remaining in crops would be harmful to human if it is applied in crop field?
2. In previous and current papers, CsA functions as a insecticide by inhibiting calcineurin activity. How about direct way to decrease or inhibit CaN to destroy insect immune system?
3. The authors observed the synergistic effects in the combinations of CsA with indoxacarb, emamectin benzoate and Vip3Aa against FAW larvae. But the authors only described the roles of indoxacarb, emamectin benzoate and Vip3Aa. It is better to discuss why CsA enhance their toxicity on insects.
Author Response
Point 1: As mentioned in the paper, the authors used high doses (100ug/g) of CsA to interfere FAW growth and development. Do authors consider the CsA remaining in crops would be harmful to human if it is applied in crop field?
Response 1: As your reminded, we had considered the safety of CsA to human if it was used in the field when we conducted this experiment. As we all known, the saft of a medicine or a pesticide is always considered in the dose.
Firstly, we used 100 ug/g of CsA to interfere third instar FAW growth and development, but the LC50 = 9.69 μg/g for newly hatched larvae, we believe that the concentration of CsA will be lower than that used in the field because the suitable methods and formulations.
Secondly, CsA was approved by the Food and Drug Administration in 1983 (Kolata, 1983) and was defined as a natural product (World Health Organization, 2009). Recent studies also confirmed that CsA was safe to humans, and even for pregnant women (Wang et al., 2021). CsA dose used in clinical application is from 3 to 25 mg/kg per day (World Health Organization, 2009), which indicates that a 60 kg adult should eat at least 180 mg every day for several days or weeks to achieve the purpose of treatment. This dose is much larger than that used in field. We discussed this question in updated manuscript.
References:
Kolata, G. FDA speeds approval of cyclosporin. Science. 1983, 221, 1273. https://doi.org/10.1126/science.221.4617.1273-aWang, N.; Ge, H.; Zhou, S. Cyclosporine A to treat unexplained recurrent spontaneous abortions: A prospective, randomized, double-blind, placebo-controlled, single-center trial. Int. J. Women's Health. 2021, 13, 1243–1250. https://doi.org/10.2147/IJWH.S330921
World Health Organization, Stuart, M.C.; Kouimtzi, M.; Hill, S.R. eds. (2009). WHO model formulary 2008. https://apps.who.int/iris/handle/10665/44053
Point 2: In previous and current papers, CsA functions as a insecticide by inhibiting calcineurin activity. How about direct way to decrease or inhibit CaN to destroy insect immune system?
Response 2: In fact, we firstly reported it CsA functions as an insecticide by inhibiting calcineurin activity. Our conclusion based on the following reasons:
1: In humans, the main role of CsA is to form a specific complex with cyclophilin that binds calcineurin (CaN), thus preventing lymphocyte proliferation and the transcription of lymphocyte factors (tumour necrosis factor alpha, interleukin 2, and interferon gamma) by inhibiting serine threonine protein phosphatase activity; this process ultimately leads to immunosuppression (Archer et al. 2018).
2. The amino acid sequences of CaNs and cyclophilin in different species are conserved (Chen and Zhang, 2013; Zhao et al. 2018; Figure R1 R2), which indicate that the action model of CsA inhibiting CaN activity in insect is similar to that in humans. CaN may plays similar functions in insects.
3. Previous studies in Drosophila and H. armigera showed that CaN activity can be increased by gram-negative bacteria to promote the product of relish, a key transcription factor of the immune deficiency (IMD) pathway in the innate immunity, which finally regulates antimicrobial peptide expression. (Dijkers and O’Farrell, 2007; Li and Dijkers, 2015; Wei et al. 2019).
4. Our experiment data showed CsA inhibited calcineurin (CaN) activity in Spodoptera frugiperda , and it showed the significantly lethal and sublethal insecticidal acticity.
However, anbout the reviewer’s question (How about direct way to decrease or inhibit CaN to destroy insect immune system), we could not provide an accurate answer. As there is a few studies reported the mechanism of CsA inhibiting CaN activity of insects and the role of CaN plays in insect immune system is still not fully understood. It will be the key job in our next work.
References:
Chen, X, Zhang, Y. (2013). Molecular cloning and characterization of the calcineurin subunit A from Plutella xylostella. Int. J. Mol. Sci. 14, 20692–20703. https://doi.org/10.3390/ijms141020692
Zhao, W, Li, L, Zhang, Y, Liu, X, Wei, J, Xie, Y, Du M, An S. (2018). Calcineurin is required for male sex pheromone biosynthesis and female acceptance. Insect Mol. Biol. 27, 373–382. https://doi.org/10.1111/imb.12379
Dijkers, PF, O’Farrell, PH. (2007). Drosophila calcineurin promotes induction of innate immune responses. Curr. Biol. 17, 2087–2093. https://doi.org/10.1016/j.cub.2007.11.001
Li, Y, Dijkers, PF, (2015). Specific calcineurin isoforms are involved in Drosophila Toll immune signaling. J. Immunol. 194, 168–176. https://doi.org/10.4049/jimmunol.1401080
Wei J, Li L, Yao S, Yang S, Zhou S, Liu X, Du M, An S (2019). Calcineurin-modulated antimicrobial peptide expression is required for the development of Helicoverpa armigera. Front Physiol. 10:1312. https://doi.org/10.3389/fphys.2019.01312
Point 3: The authors observed the synergistic effects in the combinations of CsA with indoxacarb, emamectin benzoate and Vip3Aa against FAW larvae. But the authors only described the roles of indoxacarb, emamectin benzoate and Vip3Aa. It is better to discuss why CsA enhance their toxicity on insects.
Response 3: We have discussed some possible reasons why CsA enhance the toxicity of indoxacarb, emamectin benzoate, and Vip3Aa against S. frugiperda in updated manuscript as followed:
Among the insecticides combined with CsA in this study, indoxacarb is bioactivated by esterase to prevent the entrance of Na+ into nerve cells and block nerve conduction (Wing et al. 2000). Emamectin benzoate causes the indraft of Cl-, which is an allosteric modulator of the γ-aminobutyric acid receptor (GABA) and glutamate-gated chloride channel (Glu-Cl), in nerve cells to prevent muscle contractions (Park et al. 2018). Differ from them, CsA binds to cyclophilin and inhibits CaN activity by increasing Ca2+ concentrations after CsA enters the cells (Patel and Wairkar 2019). This action model may cause synergistic combined toxicity of indoxacarb/ emamectin benzoate + CsA. Because intracellular Ca2+ rise not only render sodium channels more sen-sitive to indoxacarb but activate chloride channel (Caballero et al. 2019; Hartzell et al. 2005). Vip3Aa mainly acts on the receptors of brush border membrane vesicles, result-ing in the swelling and lysing of midgut epithelium cells (Boukedi et al. 2015). CsA may inhibit immuno-reaction of FAW larvae to cause an increase in amount of the midgut microbiota, leading to more larvae died from septicemia. (Line 224-237)
Reference:
Patel, D.; Wairkar, S. Recent advances in cyclosporine drug delivery: challenges and opportunities. Drug Deliv. Transl. Res. 2019, 9, 1067–1081. https://doi.org/10.1007/s13346-019-00650-1
Wing, K.D.; Sacher, M.; Kagaya, Y.; Tsurubuchi, Y.; Mulderig, L.; Connair, M.; Schnee, M. Bioactivation and mode of action of the oxadiazine indoxacarb in insects. Crop Prot. 2000, 19, 537–545. https://doi.org/10.1016/s0261-2194(00)00070-3
Park, J.M. Rapid development of life-threatening emamectin benzoate poisoning. Emerg. Med. 2018, 50, 81–84. https://doi.org/10.12788/emed.2018.0084
Caballero, J. P.; Murillo, L.; List, O.; Bastiat, G.; Flochlay-Sigognault, A.; Guerino, F., Apaire-Marchais, V. (2019). Nanoen-capsulated deltamethrin as synergistic agent potentiates insecticide effect of indoxacarb through an unusual neuronal cal-cium-dependent mechanism. Pestic Biochem Physiol. 2019, 157, 1-12. https://doi.org/10.1016/j.pestbp.2019.03.014
Hartzell, C.; Putzier, I.; Arreola, J. Calcium-activated chloride channels. Annu. Rev. Physiol. 2005, 67, 719-758. https://doi.org/10.1146/annurev.physiol.67.032003.154341
Boukedi, H.; Ben, Khedher, S.; Triki, N.; Kamoun, F.; Saadaoui, I.; Chakroun, M.; Tounsi, S.; Abdelkefi-Mesrati, L. Overpro-duction of the Bacillus thuringiensis Vip3Aa16 toxin and study of its insecticidal activity against the carob moth Ectomyelois ceratoniae. J. Invertebr. Pathol. 2015, 127, 127–129. https://doi.org/10.1016/j.jip.2015.03.013
Reviewer 2 Report
These are my main comments on the manuscript (toxins-1912675) entitled “Cyclosporin A acts as a novel source of insecticidal product for controlling in Spodoptera frugiperda”. The manuscript investigates the lethal and sublethal caused by Cyclosporin A as a novel insecticide against S. frugiperda. Following substantial revisions should be incorporated in the manuscript prior to acceptance.
1. I have concerns about the manuscript sections that I believe need to be addressed in order to improve its clarity.
2. A hypothesis for this work is needed.
3. Information about sublethal effects parameters are missing in methods section.
4. Other revisions could be checked in PDF attached.

Author Response
Point 1: I have concerns about the manuscript sections that I believe need to be addressed in order to improve its clarity.
Response 1: We have revised language, logic, and added some necessary descriptions in manuscript to improve its clarity.
Point 2: A hypothesis for this work is needed.
Response 2: A hypothesis for this work has been added in “Introduction” section of updated manuscript (Line 66,67)).
Point 3: Information about sublethal effects parameters are missing in methods section.
Response 3: We have added information about sublethal effects parameters in “Materials and Methods” section of updated manuscript (Line 346).
Point 4: Other revisions could be checked in PDF attached.
Response 4: We have revised manuscript according revisions mentioned in PDF attached.
Reviewer 3 Report
General comments:
The manuscript “ Cyclosporin A acts as a novel source of insecticidal product for 2 controlling in Spodoptera frugiperda” provides an interesting study of the activity of a fungal metabolite on acute and sublethal toxicity to an important crop pest. The authors have completed many experiments to better understand the effects of CsA to the fall army worm, but in many cases have not fully analyzed the data they present (larval and pupal weights) that are then described in the discussion section. The manuscript would benefit from information on the potential risk to non-target species such as beneficial insects (since it is promoted for use in integrated pest management) and humans (since CsA is a commonly used medication with immune modulating activity). Further improvements can be made to English language and grammar before the manuscript can be considered for publication. Please find specific suggestions and comments in the section below.
Specific comments:
Abstract
Line 6 – Edit “its invasive traits and severe insecticide resistance” to “its wide host plant range and reported resistance to many insecticides”
Line 9 – Add “The” before “present study”
Introduction
Line 24 – Edit “metabolism” to “metabolite”
Line 26 – Edit “ could be product by many fungi” to “could be produced by many fungi”
Line 28 - Edit “ study of action mechanism shows that CsA serves as” to “mode of action of CsA is as”
Line 30 – Delete “in life science”
Line 45 – Add “crop” before “pests”
Line 46 – Add authority after “Spodoptera frugiperda”
Lines 47-48 – Replace “agriculture” with “crop”
Line 53 – Replace “has almost invaded all over the world due to its migration habit” with “ is considered a global pest”
Line 55 – Edit “resistances to more than 40 chemical pesticides” to “resistance to more than 40 chemical insecticides”
Lines 66-67 – Edit “lepidopteran pests and also provide a new clue or mechanism for the discovery of new 66 pesticides for pest control” to “this lepidopteran pest with a new insecticidal mechanism of action”
Materials and Methods
Line 300 – What was the FAW reared on (plant or diet)?
Line 313 – How were the larvae tested or examined when “regarded as death” – please provide more details
Line 317 – Edit “subjected to dissect the midguts” and “CaN activities, respectively” to “ the midguts were dissected” and “CaN activity”
Line 323 – Please provide full details on the CaN bioassay
Lines 327-330 – Where were the insecticides obtained (company, city, country etc)? Please add more details.
Line 348 – Please provide more details on the observations made for sublethal toxicity
Results
Lines 71-74 – The information is available in Table 1, no need to repeat all in the text (could leave out the confidence limits)
Table 1 – Edit “newly hatched” with “1st instar”
Figure 3 – No statistics were completed on Fig 3A and B results (length, weight of larvae and number in each larval stage)
Figure 4 – Fig 4A – Edit axis label “Pupation rate” to “Percent of larva to pupate”; Fig 4B and C – no statistics completed on data; Fig 4D axis label – Edit “Malformation rate” to “Percent of malformed pupae”
Figure 5 – Fig 5B – no statistics on adult size
Discussion
Lines 258-259 – The larva and pupa weights were not weighed or analyzed in results section – the authors will need to include this information if they want to include as part of the discussion
The authors should include information about the risk of using CsA from the perspective of exposure to humans (it is a drug that affects immune system) and non-target effects to beneficial insects (since it is promoted as part of IPM)
Author Response
Reviewer 3
Point 1: The manuscript “ Cyclosporin A acts as a novel source of insecticidal product for 2 controlling in Spodoptera frugiperda” provides an interesting study of the activity of a fungal metabolite on acute and sublethal toxicity to an important crop pest.
Response 1: Thanks for your positive feedback.
Point 2: The authors have completed many experiments to better understand the effects of CsA to the fall army worm, but in many cases have not fully analyzed the data they present (larval and pupal weights) that are then described in the discussion section.
Response 2: We analyzed larval and pupal weights, which were showed in Figure S1, S2B, S2C in Supplementary Materials.
Point 3: The manuscript would benefit from information on the potential risk to non-target species such as beneficial insects (since it is promoted for use in integrated pest management) and humans (since CsA is a commonly used medication with immune modulating activity).
Response 3: We have discussed the risk of using CsA from the perspective of exposure to humans and beneficial insects in section “discussion” of updated manuscript.
Point 4: Further improvements can be made to English language and grammar before the manuscript can be considered for publication. Please find specific suggestions and comments in the section below.
Response 4: We sent our manuscript to a native English speaker for revision. The language and grammar have been improved according to revision.
Specific comments:
Point 5: Abstract
Line 6 – Edit “its invasive traits and severe insecticide resistance” to “its wide host plant range and reported resistance to many insecticides”
Line 9 – Add “The” before “present study”
Introduction
Line 24 – Edit “metabolism” to “metabolite”
Line 26 – Edit “ could be product by many fungi” to “could be produced by many fungi”
Line 28 - Edit “ study of action mechanism shows that CsA serves as” to “mode of action of CsA is as”
Line 30 – Delete “in life science”
Line 45 – Add “crop” before “pests”
Line 46 – Add authority after “Spodoptera frugiperda”
Lines 47-48 – Replace “agriculture” with “crop”
Line 53 – Replace “has almost invaded all over the world due to its migration habit” with “ is considered a global pest”
Line 55 – Edit “resistances to more than 40 chemical pesticides” to “resistance to more than 40 chemical insecticides”
Lines 66-67 – Edit “lepidopteran pests and also provide a new clue or mechanism for the discovery of new 66 pesticides for pest control” to “this lepidopteran pest with a new insecticidal mechanism of action”
Response 5: We have edited these sections of manuscript according to reviewer’s suggestion.
Materials and Methods
Point 6: Line 300 – What was the FAW reared on (plant or diet)?
Response 6: The newly hatched larvae were fed fresh corn leaves in plastic boxes until they developed to the 3rd-instar stage. To prevent cannibalism among larvae, 3rd-instar larvae were reared individually with approximately 0.5 g of artificial diet (Liang et al., 1999) in 25 mL cups. The artificial diet was replaced daily until pupation. The sexes of newly-emerged adults were distinguished according to the external genitalia at the end of the abdomen. The adults were reared in cages with 10% sucrose solution for subsequent mating and reproduction. We described the method in supplementary methods and materials of Supplementary Materials.
Point 7: Line 313 – How were the larvae tested or examined when “regarded as death” – please provide more details
Response 7: Larvae who have no response to physical stimulation, and larvae whose instar remained unchanged after 7 days were recorded as death.
Poing 8: Line 317 – Edit “subjected to dissect the midguts” and “CaN activities, respectively” to “ the midguts were dissected” and “CaN activity”
Response 8: We have edited according suggestion.
Point 9: Line 323 – Please provide full details on the CaN bioassay
Response 9: Briefly, midgut tissue proteins and CaN assay buffer were mixed, and CaN substrate was added and incubated for 10 min. Next, color development reagent was added and the (OD) 636 nm was read by the SynergyH1 Hybrid Multi-Mode Reader (BioTek, Vermont, USA). Three replicates were conducted. CaN activity was calculated following the formula provided by the manufacturer’s instructions. We added this description in section 5.4.
We think that our description is enough for replication of CaN bioassay, and the manufacturer’s instructions of BCA Protein Assay Kit (Beyotime Biotechnology, Shanghai, China) and Calcineurin Activity Assay Kit (Abcam, Cambridge, UK) are easily obtained.
Point 10: Lines 327-330 – Where were the insecticides obtained (company, city, country etc)? Please add more details.
Response 10: We have transferred information of insecticides from Supplementary Materials to section 2.2 of “Materials and Methods”.
CsA (98.5% active ingredient) was purchased from Beijing Solarbio Science and Technology Co., Ltd. (Beijing, China). Indoxacarb (99% active ingredient), chlorantraniliprole (95% active ingredient), and emamectin benzoate (91% active ingredi-ent) were kindly provided by Dr Bin Zhu (Department of Entomology, China Agricultural University, Beijing, China). Vip3Aa protein was purchased from Beijing Genralpest Bio-tech Research Co., Ltd (Beijing, China)
Point 11: Line 348 – Please provide more details on the observations made for sublethal toxicity
Response 11: The details on the observations made for sublethal toxicity were described in Supplementary Materials as followed:
Bioassay of sublethal CsA doses against FAW
Third-instar larvae were fed an artificial diet containing different doses of CsA, and all live larvae in each treatment (120 3rd-instar larvae of 3 replicates) were weighed every 2 days. At 7 days after CsA application, the numbers of larvae in different instar stages were recorded (3rd-instar larvae were recorded as dead). The larval development stages were distinguished by molting time. Larvae in the prepupal stage were observed every 12 h to determine the time of successful pupation. Thus, the development period was recorded from the 3rd-instar stage until successful pupation. Additionally, the numbers of larvae in different instar stages were recorded after 8 days of treatment (360 larvae of 3 replicates for each treatment).
For each treatment, pupation rates were calculated by dividing the number of pupae (including abnormal pupae) by the number of larvae (120 individuals for each replicate). Malformation rates were calculated by dividing the number of abnormal pupae by the total number of all pupae. Female and male pupae were distinguished according to their abdomen characteristics for analysis of the female/male ratio. Female and male pupae (including malformed pupae) were weighed 1 day after pupation. The pupal stage was defined as the number of days from pupae to successful emergence.
All adults received a 10% sucrose solution without CsA or DMSO every daily. Newly-emerged adults in each treatment group were transferred singly to a 25 mL cup. Eclosion rates were obtained by dividing the number of successfully emerged adults by the number of normal pupae. The adult period (excluding mating) was defined as the number of days from successful adult emergence until death. At least 27 female adults of 3 replicates were used to investigate the sublethal effects of CsA on ovarian development. The number of mature eggs in ovaries (eggs between the ovipositor and pink parts of the oviducts; Fig. 5C) and ovarian length were recorded for 5-day-old virgin female adults. In the mating survey, vigorous female and male adults in each treatment (at least 19 pairs for each replicate) were selected on the 4th-day after emergence and placed singly in a 450 mL box covered with a piece of cotton gauze. After 1 day of mating, female adults with hard spermatophores were considered to have mated. To investigate the effects of CsA on FAW fecundity, a pair of 3-day-old adults were placed singly in a box, and the female was observed every morning; the date when the female began to lay eggs was recorded. Once the female began to lay eggs on the cotton gauze and box wall, the pair of adults was transferred to a new box covered with a piece of cotton gauze every day to facilitate egg counting; the eggs were collected into a sealed plastic bag with sufficient moisture. The hatching rate was analyzed according by the number of newly hatched larvae and number of eggs laid by females.
Larvae (8 days after CsA application), pupae (2 days after pupation) and adults (1 day after emergence) in the different treatments were observed using a camera (EOS R, Canon, Tokyo Japan). Ovaries were photographed using an anatomical microscope (M205 A; Leica, Welzlar, Germany).
Results
Point 12: Lines 71-74 – The information is available in Table 1, no need to repeat all in the text (could leave out the confidence limits)
Response 12: The data of confidence limits have been deleted in the text.
Point 13: Table 1 – Edit “newly hatched” with “1st instar”
Response 13: “First instar” has been replaced “newly hatched.”
Point 14: Figure 3 – No statistics were completed on Fig 3A and B results (length, weight of larvae and number in each larval stage)
Response 14: Data of larval weight is showed in Figure S1 in Supplementary Materials. Number in each larval stage were added in section 2.4 of manuscript. Indeed, photographs of larval bodies are mainly used to show the larval phenotypes caused by CsA. We think that larval weight and periods affected by CsA are enough to indicate negative effects caused by CsA, therefore we did not record larval length. We added a measuring scale in Figure 3A.
Point 15: Figure 4 – Fig 4A – Edit axis label “Pupation rate” to “Percent of larva to pupate”; Fig 4B and C – no statistics completed on data; Fig 4D axis label – Edit “Malformation rate” to “Percent of malformed pupae”
Response 15: We revised axis label of Figure 4A and Figure 4D according suggestion. Similar to larvae, we think pupal weights is enough to indicate negative effects caused by CsA. Figure 4A and Figure 4C were mainly used to show the phenotypes caused by CsA. We also respectively added a measuring scale in Figure 4A and Figure 4D.
Point 16: Figure 5 – Fig 5B – no statistics on adult size
Response 16: We added a measuring scale in Figure 5B. Adult sizes were mainly used to indicate that suppression of ovarian development is greater than that of body size.
Discussion
Point 17: Lines 258-259 – The larva and pupa weights were not weighed or analyzed in results section – the authors will need to include this information if they want to include as part of the discussion
Response 17: Indeed, we weighed and analyzed larval and pupal weights. The data were showed in Figure S1 and Figure S2B, S2C in Supplementary Materials.
Point 18: The authors should include information about the risk of using CsA from the perspective of exposure to humans (it is a drug that affects immune system) and non-target effects to beneficial insects (since it is promoted as part of IPM)
Response 18: We have discussed the risk of using CsA from the perspective of exposure to humans and beneficial insects in section “discussion” of updated manuscript. The information is as followed:
The risk of insecticides to humans and beneficial insects is very important to evalu-ating their safety and toxicity. CsA has been approved by the Food and Drug Administra-tion in 1983 (Kolata, 1983). Recent studies also confirmed that CsA is safe to humans, and even for pregnant women (Wang et al., 2021). CsA dose used in clinical application is from 3 to 25 mg/kg per day (World Health Or-ganization, 2009), which may be great larger than the dose used for insect control. For beneficial insects, like wasp parasitoids may benefit from CsA application to improve population. Because parasitic ability of wasp parasitoids is closely related to the immune response of host insects (Cerenius et al., 2010; Asgari et al., 2011). Eggs and larvae of wasp parasitoids may more easily survive in host insects whose immunoreaction has been inhibited by CsA. Therefore, the security evalua-tion of CsA to beneficial insects needed to be roundly perform in next work.
References:
Kolata, G. FDA speeds approval of cyclosporin. Science. 1983, 221, 1273. https://doi.org/10.1126/science.221.4617.1273-a
Wang, N.; Ge, H.; Zhou, S. Cyclosporine A to treat unexplained recurrent spontaneous abortions: A prospective, randomized, double-blind, placebo-controlled, single-center trial. Int. J. Women's Health. 2021, 13, 1243–1250. https://doi.org/10.2147/IJWH.S330921
World Health Organization, Stuart, M.C.; Kouimtzi, M.; Hill, S.R. eds. (2009). WHO model formulary 2008. https://apps.who.int/iris/handle/10665/44053
Cerenius, L.; Kawabata, S.; Lee, B.L.; Nonaka, M.; Söderhäll, K. Proteolytic cascades and their involvement in invertebrate immunity. Trends Biochem. Sci. 2010, 35, 575-83. https://doi.org/10.1016/j.tibs.2010.04.006
Asgari, S.; Rivers, D.B. Venom proteins from endoparasitoid wasps and their role in host-parasite interactions. Annu. Rev. Entomol. 2011, 56, 313-35. https://doi.org/10.1146/annurev-ento-120709-144849
Round 2
Reviewer 2 Report
The authors have incorporated all suggestions and comments into the revised version, now the manuscript seems much clear. There is minor point to be corrected:
Ls.35, 38, 194, 260, and 266: For these arthropod species, provide the ID author scientific name, order and family taxa.
L.65: The median and 95% lethal concentrations…
Table 1: Please, revise the p-value. In probit analysis, value should be p>0.05.
L.95: … synergistic toxicity….
Ls.110-114: Revise this sentence to eliminate rewordiness.
Ls.125-137: In addition, A brief description of the malformations found in the pupae is needed.
L.243: In Lepidoptera,…
L.244: …larval development…
L.331: Pull apart “previously[24,25]”
Author Response
Point 1: The authors have incorporated all suggestions and comments into the revised version, now the manuscript seems much clear. There is minor point to be corrected:
Response 1: Thanks for your nice suggestions. Green highlight was used to mark the revision.
Point 2: Ls.35, 38, 194, 260, and 266: For these arthropod species, provide the ID author scientific name, order and family taxa.
Response 2: The ID author scientific name, order and family taxa were added in updated MS. Please see lines 33, 34, 35, 38, 194, 265, and 271.
Point 3:L.65: The median and 95% lethal concentrations…
Response 3: We have replaced “lethal doses” with “lethal concentrations”. Please see line 65.
Point 4:Table 1: Please, revise the p-value. In probit analysis, value should be p>0.05.
Response 4: The p-value are correct. In probit analysis, heterogeneity would be used in the calculation of confidence limits when p < 0.15.
Point 5:L.95: … synergistic toxicity….
Response 5: “synergistic toxicityt” has been replaced with “synergistic toxicity”, please see line 97.
Point 6:Ls.110-114: Revise this sentence to eliminate rewordiness.
Response 6: We have revised this sentence. Please see line 108-113.
Point 7: Ls.125-137: In addition, A brief description of the malformations found in the pupae is needed.
Response 7:Indeed,a brief description of the malformations found in pupae has been described in MS. Please see 127-128.
Point 8:L.243: In Lepidoptera,…
Response 8: We have replaced “lepidoptera” with “Lepidoptera”. Please see line 247.
Point 9:L.244: …larval development…
Response 9: We have replaced “larval transformation” with “larval development”. Please see line 248.
Point 10:L.331: Pull apart “previously[24,25]”
Response 10: A space was added. Please see line 336.
Reviewer 3 Report
The reviewer appreciates the revisions the authors have made to the manuscript, and it has been improved from the first version. However there remains several improvements that should be made prior to publication. The In terms of the English language, there are still many instances where the sentence structure and grammar could be improved - for example in the highlighted section in the discussion: " The risk of insecticides to humans and beneficial insects is very important to evaluating their safety and toxicity. CsA has been approved by the Food and Drug Administra-tion in 1983 [28].- change to "The toxicity of insecticides to humans and beneficial insects is very important to evaluate even though CsA has been approved by the Food and Drug Administration in 1983 [28]." Please ensure these types of mistakes are corrected throughout the text.
The authors made most of the suggested changes, but there were several that were not completed or perhaps misunderstood:
Response 5: We have edited these sections of manuscript according to reviewer’s suggestion.
Appears that a sentence is missing where these edits were supposed to be – in Lines 22-24
“crop” should be pluralized to “crops” in Line 43
Response 7: Larvae who have no response to physical stimulation, and larvae whose instar remained unchanged after 7 days were recorded as death.
Edit to read “recorded as dead” – in Line 307
Response 11: The details on the observations made for sublethal toxicity were described in Supplementary Materials as followed:
"For each treatment, pupation rates were calculated by dividing the number of pupae (including abnormal pupae) by the number of larvae (120 individuals for each replicate). Malformation rates were calculated by dividing the number of abnormal pupae by the total number of all pupae."
Rates are based on unit of time – what is described for pupation and malformation represent a percentage or proportion of each
Response 14: Data of larval weight is showed in Figure S1 in Supplementary Materials. Number in each larval stage were added in section 2.4 of manuscript. Indeed, photographs of larval bodies are mainly used to show the larval phenotypes caused by CsA. We think that larval weight and periods affected by CsA are enough to indicate negative effects caused by CsA, therefore we did not record larval length. We added a measuring scale in Figure 3A.
Not necessary to add the number of larvae in the text as it is shown in Fig 3B; the measuring scale should be parallel to the larvae rather than perpendicular
Response 15: We revised axis label of Figure 4A and Figure 4D according suggestion. Similar to larvae, we think pupal weights is enough to indicate negative effects caused by CsA. Figure 4A and Figure 4C were mainly used to show the phenotypes caused by CsA. We also respectively added a measuring scale in Figure 4A and Figure 4D.
As with Fig 3, the scale should be parallel to the insect body length
Response 16: We added a measuring scale in Figure 5B. Adult sizes were mainly used to indicate that suppression of ovarian development is greater than that of body size.
Same as with Fig 3 and 4
Response 18: We have discussed the risk of using CsA from the perspective of exposure to humans and beneficial insects in section “discussion” of updated manuscript. The information is as followed:
Authors should indicate that exposure to CsA would be different for humans through pesticide application (inhalation or contact) versus clinic application (ingestion) so the effects of CsA on humans are not clear. It might be better for the authors to indicate that further toxicological tests are required for humans, mammals and other non-target species.
Author Response
Point 1: The reviewer appreciates the revisions the authors have made to the manuscript, and it has been improved from the first version. However there remains several improvements that should be made prior to publication. The In terms of the English language, there are still many instances where the sentence structure and grammar could be improved - for example in the highlighted section in the discussion: " The risk of insecticides to humans and beneficial insects is very important to evaluating their safety and toxicity. CsA has been approved by the Food and Drug Administra-tion in 1983 [28].- change to "The toxicity of insecticides to humans and beneficial insects is very important to evaluate even though CsA has been approved by the Food and Drug Administration in 1983 [28]." Please ensure these types of mistakes are corrected throughout the text.
Response 1: We have revised our MS according to your nice suggestions. Blue highlight was used to mark them.
The authors made most of the suggested changes, but there were several that were not completed or perhaps misunderstood:
Point 2: Response 5: We have edited these sections of manuscript according to reviewer’s suggestion.
Appears that a sentence is missing where these edits were supposed to be – in Lines 22-24
“crop” should be pluralized to “crops” in Line 43
Response 2: We have added the missing sentence. “Crop” has been replaced with “crops”. Please see line 22, 23, and 43.
Point 3: Response 7: Larvae who have no response to physical stimulation, and larvae whose instar remained unchanged after 7 days were recorded as death.
Edit to read “recorded as dead” – in Line 307
Response: We have replaced “death” with “dead”. Please see line 311.
Point 4: Response 11: The details on the observations made for sublethal toxicity were described in Supplementary Materials as followed:
"For each treatment, pupation rates were calculated by dividing the number of pupae (including abnormal pupae) by the number of larvae (120 individuals for each replicate). Malformation rates were calculated by dividing the number of abnormal pupae by the total number of all pupae."
Rates are based on unit of time – what is described for pupation and malformation represent a percentage or proportion of each
Response 4: We did not calculate the percentage of pupation and malformation day by day. When all larvae of each group became pupae or died, the number of total pupae and abnormal pupae of each group were recorded and used to calculate the percentage of pupation and malformation.
Point 5: Response 14: Data of larval weight is showed in Figure S1 in Supplementary Materials. Number in each larval stage were added in section 2.4 of manuscript. Indeed, photographs of larval bodies are mainly used to show the larval phenotypes caused by CsA. We think that larval weight and periods affected by CsA are enough to indicate negative effects caused by CsA, therefore we did not record larval length. We added a measuring scale in Figure 3A.
Not necessary to add the number of larvae in the text as it is shown in Fig 3B; the measuring scale should be parallel to the larvae rather than perpendicular
Response: We have deleted the number of larvae. The measuring scale has been parallel to the larvae in updated MS. Please see line 108-113, 115.
Point 6: Response 15: We revised axis label of Figure 4A and Figure 4D according suggestion. Similar to larvae, we think pupal weights is enough to indicate negative effects caused by CsA. Figure 4A and Figure 4C were mainly used to show the phenotypes caused by CsA. We also respectively added a measuring scale in Figure 4A and Figure 4D.
As with Fig 3, the scale should be parallel to the insect body length
Response 6: The scale has been changed. Please see line 134.
Point 7: Response 16: We added a measuring scale in Figure 5B. Adult sizes were mainly used to indicate that suppression of ovarian development is greater than that of body size.
Same as with Fig 3 and 4
Response 7: The scale has been changed. Please see line 151.
Point 8: Response 18: We have discussed the risk of using CsA from the perspective of exposure to humans and beneficial insects in section “discussion” of updated manuscript. The information is as followed:
Authors should indicate that exposure to CsA would be different for humans through pesticide application (inhalation or contact) versus clinic application (ingestion) so the effects of CsA on humans are not clear. It might be better for the authors to indicate that further toxicological tests are required for humans, mammals and other non-target species.
Response 8: we have revised the discussion, please see line: 204-212.